# Sequestration of Drugs from Biomolecular and Biomimicking Environments: Spectroscopic and Calorimetric Studies

Rahul Yadav [1], Bijan Kumar Paul [2,*] and Saptarshi Mukherjee [1,*]

1 Department of Chemistry, Indian Institute of Science Education and Research Bhopal, Bhopal Bypass Road, Bhauri, Bhopal 462 066, Madhya Pradesh, India; rahuly19@iiserb.ac.in
2 Department of Chemistry, Mahadevananda Mahavidyalaya, Barrackpore, Kolkata 700 120, West Bengal, India
* Correspondence: paulbk.chemistry@gmail.com (B.K.P.); saptarshi@iiserb.ac.in (S.M.)

**Abstract:** The binding of drugs to nucleic acids, proteins, lipids, amino acids, and other biological receptors is necessary for the transportation of drugs. However, various side effects may also originate if the bound drug molecules are not dissociated from the carrier, especially with the aid of non-toxic agents. The sequestration of small drug molecules bound to biomolecules is thus central to counter issues related to drug overdose and drug detoxification. In this article, we aim to present several methods used for the dissociation of small drug molecules bound to different biological and biomimicking assemblies under in vitro experimental conditions. To this effect, the application of various molecular assemblies, like micelles, mixed micelles, molecular containers, like β-cyclodextrin, cucurbit[7]uril hydrate, etc., has been discussed. Herein, we also try to shed light on the driving forces underlying such sequestration processes through spectroscopic and calorimetric techniques.

**Keywords:** drug sequestration; self-assemblies; host–guest chemistry; steady-state spectroscopy; isothermal titration calorimetry





## 1. Introduction

The study of the interaction between small molecules and biomolecules, like proteins, DNA, RNA, lipids membranes, etc. forms a key area of research in which a comprehensive understanding of such interactions lays the foundation for the requisite understanding toward the development of new drugs to counter various lethal diseases [1–3]. Various research groups have, over the years, investigated the interactions of small molecules with proteins, DNA, RNA, etc. by using experimental techniques based on spectroscopy, calorimetry, polarimetry, and so on [4–10]. In recent times, a significant body of computational or theoretical research endeavors has also been dedicated toward a better understanding of the interaction phenomena [11–17]. The increasing growth of attention, within the scientific community, to this field of research is not surprising because the site-specific targeted delivery of drug molecules and their excretion from the human body via a benign mechanism pose very crucial research questions toward the successful implementation of the therapeutic actions of a drug. The issue related to the excretion of drugs becomes even more pertinent with the view of the possibility of overdose, which may in turn lead to detrimental effects on various biomolecular assemblies within the human body [18–20]. In this context, it may be noted that in the last few years, the evolution in the field of nanomedicine has promised developments of smart systems aiming at in vivo functionality. However, it is important to realize that a large part of the promise developed in this area still relies on rational reasoning rather than concrete experimental evidence, thereby inviting further research to this effect [21,22]. Under real in vivo conditions, nanocarriers or nano-assemblies should first arrive at the site of interest, such as target organs/tissues, and then have the possibility of sequestering small drug molecules bound to biomolecules. However, nanocarriers or nano-assemblies themselves will always encounter issues related to nano-bio interactions. Therefore, the targeting ability becomes critical for these kinds

of concepts [21,22]. Naturally, the validation of such concepts still promises ample room for rigorous experiments. However, given the intrinsic complexities associated with the in vivo studies on such effects, it is often beneficial to explore the concerned interactions in an in vitro experimental setup to extract the initial guidelines. The key motivation of the present article is to review the application of various molecular assemblies in the sequestration of drugs bound to different biological assemblies, like DNA, RNA, proteins, etc., and biomimicking assemblies, like lipid membranes. Herein, we summarize the experimental data showing the extraction of bound drugs using self-assembled molecular aggregates, like micelles, mixed micelles, reverse micelles, liposomes, niosomes, etc., as well as using molecular receptors, like cyclodextrins, and cucurbit[n]urils. We feel that the results of the sequestration of bound-drug molecules from a variety of biological/biomimicking receptors could be helpful not only in forming a comprehensive understanding of this important phenomenon but also in furnishing a relevant database for optimizing the accompanying experimental conditions.

Micelles are self-assemblies of amphiphilic molecules having long hydrophobic tails oriented toward the core and hydrophilic headgroups exposed toward the bulk water. They are thermodynamically stable, nanoscopic dynamic structures typically formed in the aqueous medium above a certain concentration (the critical micellar concentration, CMC) and above a certain temperature (the critical micellar temperature (CMT) or the Krafft temperature) [23–25]. The surfactant-assisted dissociation of a drug–DNA complex was first reported by Muller and Crothers in 1968 [26]. Later in 2003, Westerlund and co-workers showed that the rate of dissociation of DNA-bound cationic ligands is augmented by the presence of surfactants in both monomeric and micellar forms [27]. They studied the binding interaction of four cationic complexes with DNA, suggesting that the intercalative mode of binding is operational. Further, a sodium dodecyl sulfate (SDS) surfactant was employed to interact with the cationic complexes via strong electrostatic forces in which the surfactants act as hydrophobic pockets for the sequestration of the complexes. Negatively charged SDS micelles are assumed to not interact with the DNA due to electrostatic repulsion [27]. Furthermore, the same group reported that an SDS surfactant drastically accelerates the rate of the dissociation of Ruthenium complexes from calf thymus DNA (ctDNA), with this process being entropically driven [28]. Guchhait and co-workers reported that the interaction of the biological photosensitizer harmane (HM) with herring sperm DNA (hsDNA) is governed by both intercalative and electrostatic forces, and the bound drug was subsequently deintercalated using cationic surfactant cetyltrimethyl ammonium bromide (CTAB) [29]. Patra et al. reported the anionic surfactant SDS micelle-assisted deintercalation of the cationic dye phenosafranin from DNA [30]. Using Isothermal Titration Calorimetry (ITC), they showed that the overall deintercalation process is enthalpically unfavorable but entropically favorable. The circular dichroism (CD) spectra of SDS with DNA revealed that SDS micelles do not alter the native B-form of DNA [30]. Kundu et al. explored the cationic CTAB micelle-assisted dissociation of the 3-acetyl-4-oxo-6,7-dihydro-12H-indolo[2,3-a]-quinolizine (AODIQ) drug from ctDNA. By using spectroscopic and molecular docking studies, they demonstrated that AODIQ binds to DNA at the minor groove. CTAB electrostatically binds to the polyphosphate backbone of DNA and behaves as a hydrophobic sink for DNA-bound drugs [31]. Apart from cationic and anionic surfactants, Singh and co-workers reported the application of a non-ionic surfactant, Triton-X 114, as a sequestrating agent for hs-DNA-bound safranin O [32]. Below the CMC, Triton-X 114 is unable to extract the drug molecules from DNA, but beyond its CMC, the surfactant is found to extract the DNA-intercalated safranin O. CD spectroscopy further validated that the secondary structure of DNA is not influenced by Triton-X 114 at both premicellar and micellar concentrations, thereby making the non-ionic surfactant an effective drug-sequestrating probe [32]. Different drug-sequestrating agents for various biomolecular/bio-replicating systems are tabulated in Table 1 [29–48].

**Table 1.** Different sequestrating agents for different biological and biomimicking environments.

| Serial Number | Biological/Biomimicking Environment | Small Molecule/Drug | Sequestrating Agent | Reference(s) |
|---|---|---|---|---|
| 1. | Herring Sperm DNA | Harmane | CTAB | [29] |
| 2. | Calf Thymus DNA | Phenosafranin | SDS | [30] |
| 3. | Calf Thymus DNA | 3-Acetyl-4-oxo-6,7-dihydro-12H-indolo-[2,3-a]-quinolizine | CTAB | [31] |
| 4. | Herring Sperm DNA | Safranin O | Triton X-114 | [32] |
| 5. | DNA | Mitoxantrone | SDS | [33] |
| 6. | Heparin | Mallard Blue | SDS | [34] |
| 7. | Herring Sperm DNA | Norharmane | DTAB, CTAB, TTAB | [35] |
| 8. | DNA | Ethidium Bromide | P105, F127, and P123 with SDS | [36,37] |
| 9. | DNA | Epirubicin | P123-SDS Mixed Micelles | [38] |
| 10. | BSA Protein | 9-(2-Caboxy-2-cyano)vinyl-julolidine | CTAB-P123 Mixed Micelles | [39] |
| 11. | Calf Thymus DNA | Piperine | Surface Active Ionic Liquid | [40] |
| 12. | Calf Thymus DNA | Doxorubicin | Liposomes | [41–43] |
| 13. | DMPC lipids | Phenosafranin | β-Cyclodextrin | [44] |
| 14. | DMPG Lipids | Nile Red | β-Cyclodextrin | [45] |
| 15. | EYPC Lipids | Phenosafranin, ANS, Nile Red | β-Cyclodextrin | [46] |
| 16. | Liposomes | Sanguinarine | β-Cyclodextrin | [47] |
| 17. | Torula Yeast RNA | Cryptolepine | Cucurbit[7]uril | [48] |

## 2. Sequestration of Drugs from Biomolecular Assemblies

### 2.1. Sequestration of Small Molecules from DNA Using Micelles

In this section, we will discuss the mechanism of the sequestration of Harmane (HM) bound to herring sperm DNA, using the cationic surfactant CTAB [29]. Herein, the detergent molecules conventionally act as the hydrophobic sink for the dissociated drugs. Figure 1 depicts that in the presence of the cationic surfactant CTAB, the fluorescence spectral characteristics of DNA-bound HM are qualitatively reversed in comparison to those observed during the binding interaction of the drug with DNA, along with an increase in an additional fluorescence band at ~375 nm [29] attributed to the neutral species of HM [49–56]. The modulation of the fluorescence intensity of HM (the cationic species) with added CTAB concentrations is shown in the inset of Figure 1. These results reflect the deintercalation of the bound drug from the HM–DNA complex in the presence of the surfactant. As a control experiment, the interaction of HM with the surfactant alone has also been performed with the result that the inception of saturation of the interaction of CTAB with the HM–DNA complex is observed at [CTAB] ~0.6 mM, a considerably small concentration of the surfactant at which there is essentially no interaction of HM with the surfactant alone [29].

The CTAB-induced modulation of the fluorescence profile of the HM–DNA complex is thus reasonably assigned to the dissociation of the bound drug, while the development of the fluorescence band corresponding to the neutral species of HM may be due to enhanced hydrophobicity of the medium due to the presence of the surfactant. In this context, it is pertinent to state that similar observations were not found with an anionic surfactant, namely sodium dodecyl sulfate (SDS) [29]. This is probably due to the lack of an interaction of SDS with DNA owing to the electrostatic repulsion between the negatively charged polyphosphate DNA backbone and the anionic surfactant [26,27,57–63]. Similarly, with a neutral surfactant (Triton X-100), no noticeable sequestration of the DNA-bound drug molecules could be observed [29]. These observations collectively lead to the conclusion that a stabilizing electrostatic interaction between the cationic surfactant CTAB and the negatively charged polyphosphate backbone of DNA probably plays an important role underlying the observed interactions leading to the sequestration of the bound drug molecules from the DNA duplex. It is relevant to state in this context that in the range of CTAB concentrations employed in the present study, the native conformation (B-form) of DNA remains practically unperturbed, as is characterized by the circular dichroic (CD) profile of DNA with added CTAB, which resembles the CD profile of the B-DNA [29].

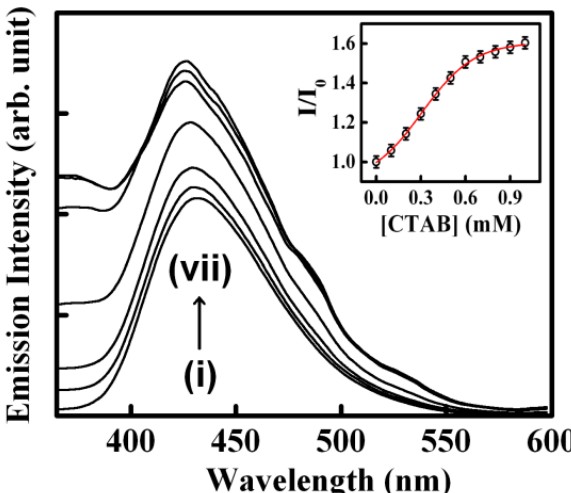

**Figure 1.** Effect of CTAB on the emission of DNA-bound HM. Spectra (i) to (vii) correspond to [CTAB] in mM (i) = 0, (ii) = 0.1, (iii) = 0.2, (iv) = 0.3, (v) = 0.4, (vi) = 0.5, (vii) = 0.6. Inset variation of cationic fluorescence ($I/I_0$ at $\lambda_{em}$ = 435 nm; $I_0$ is the fluorescence intensity of DNA-bound HM in the absence of CTAB, and I is the fluorescence intensity of DNA-bound HM with an increasing concentration of CTAB). [Adapted with permission from Ref. [29] Copyright 2011 American Chemical Society].

In the study of the interaction of drug molecules with biological receptors, the kinetics of the association and dissociation of the drug is considered to have crucial diagnostic significance [26,27,58–60]. A slow rate of deintercalation from the DNA-bound state has been widely argued in the literature as an important property toward efficient therapeutic functionality [26,27]. The fluorescence decay profile of HM in the presence of DNA (at $\lambda \sim 435$ nm) represents a characteristic rate constant for the association process, as $k_a(\pm 5\%)$ = 0.0375 s$^{-1}$ (Figure 2a), while the dissociation rate constant with added CTAB is found to be $k_d(\pm 5\%)$ = 0.0098 s$^{-1}$ (Figure 2b) [29].

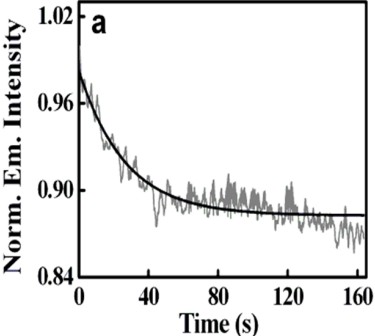
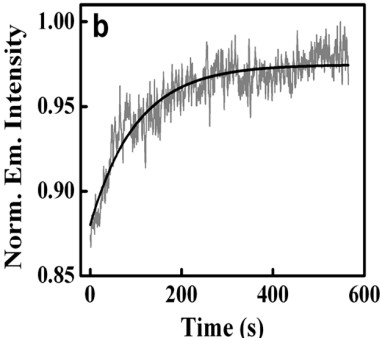

**Figure 2.** Change in the fluorescence intensity of HM at $\lambda_{em}$ = 435 nm with respect to time. (**a**) During the interaction of 2.0 µM HM with 100 µM DNA. (**b**) Dissociation of DNA-bound HM in the presence of 0.9 mM CTAB. The solid black lines indicate the fitted lines. [Adapted with permission from Ref. [29]. Copyright 2011 American Chemical Society].

However, given that the dissociation kinetic data observed in the presence of CTAB should not be directly comparable with the association kinetics data found in the absence of the surfactant, the following expression has been used to calculate a value for the dissociation rate constant:

$$K = \frac{k_1}{k_{-1}}$$

where, K = $3.34 \times 10^3$ represents the binding association constant of HM with DNA. Using this relationship, we have $k_{-1} = 1.12 \times 10^{-5}$ M$^{-1}$s$^{-1}$ [29], that is the prerequisite for a faster

association, and the relatively much slower dissociation of the drug from the DNA duplex is realized under the presently employed experimental conditions.

In another report by Paul et al. [35], the detergent-sequestered release of norharmane (NHM, Scheme 1) from the DNA double helix was addressed by using a series of cationic surfactants, namely cetyltrimethyl ammonium bromide (CTAB), tetradecyl trimethyl ammonium bromide (TTAB), and dodecyl trimethyl ammonium bromide (DTAB), with varying hydrophobic tail lengths (n = 16 for CTAB, n = 14 for TTAB and n = 12 for DTAB; n = number of carbon atoms in the surfactant tail length).

**Scheme 1.** The cation ⇌ neutral prototropic equilibrium of NHM. [Reprinted with permission from Ref. [35]. Copyright 2016, Elsevier].

The observations (Figure 3) of cationic surfactant-induced modulations in the fluorescence profile of DNA-bound NHM in a qualitatively reverse pattern in comparison to those observed for the binding of the drug to the DNA duplex is ascribed to the deintercalation of the drug from the DNA scaffolds, whereas the genesis of the additional fluorescence band at ~380 nm is attributed to the neutral species of the drug [29,35,64–66]. The negligible interaction of NHM with the surfactants alone within the range of concentrations of the surfactants employed for deintercalation of the drug was also confirmed in suitably designed control experiments [35]. The important criterion of a slow rate of deintercalation of the drug compared to the rate of association is also achieved under the as-employed experimental conditions. The fluorescence decay traces describing the deintercalation of the DNA-bound drug in the presence of the surfactants are presented in the insets of Figure 3, and the relevant data are compiled in Table 2.

**Table 2.** Summary of kinetic parameters for the surfactant-sequestered dissociation of DNA-bound NHM at 293 K.

| Surfactant | n [a] | $k_d$ $(s^{-1})$ |
|:---:|:---:|:---:|
| CTAB | 16 | $65 \times 10^{-3}$ |
| TTAB | 14 | $12 \times 10^{-3}$ |
| DTAB | 12 | $7.96 \times 10^{-3}$ |

[a] n = no. of C-atoms in the surfactant chain. The rate constant for the association of NHM with DNA at 293 K is $k_a = 108.1 \times 10^{-3}$ $s^{-1}$. [Reprinted with permission from Ref. [35]. Copyright 2016, Elsevier].

In this context, it is important to note that simply through the rational choice of the surfactants, the rate of dissociation of the drug from the DNA scaffolds can be tuned effectively over an order of magnitude (Table 2). This in turn provides considerable flexibility in the wider perspective of research on drug delivery toward optimization of the ADME (Administration-Distribution-Metabolism-Elimination) profile of the drug, and our results appear to present a tenable initiative for this effect in an in vitro study.

*Thermodynamics of dissociation.* The thermodynamic parameters corresponding to the dissociation of the NHM–DNA complex with an added surfactant, as derived from isothermal titration calorimetry (ITC) measurements, are presented in Table 3 (and the results of ITC measurements for the interaction of CTAB with the NHM–DNA complex are displayed in Figure 4). That the native conformation of the DNA double helix is not discernibly perturbed in the presence of the surfactants within the range of concentrations of the latter implies the possibility of no direct interaction of DNA with the surfactants.

Consequently, it is argued that the enhanced hydrophobicity of the medium in the presence of the surfactants stimulates a transit opening of the compact double-helical DNA structure. This in turn provides a hydrophobic sink for deintercalation of the bound drug molecules from the drug–DNA complex by decreasing the energy of activation of the process [26,27]. The deintercalated drug molecules are subsequently associated with the surfactant molecules. The first step is typically governed by pivotal contributions from hydrophobic interactions and is usually recognized as the rate-limiting step. However, it is important to realize that the thermodynamic data derived from ITC measurements provide an equilibrium description of the inherently complex phenomenon. The observed increase in entropy ($T\Delta S > 0$) is in consensus with the typical thermodynamic signature for hydrophobic hydration in which a positive change of entropy is described based on the release of water molecules and the counterions from the interface of the interacting species [67–72].

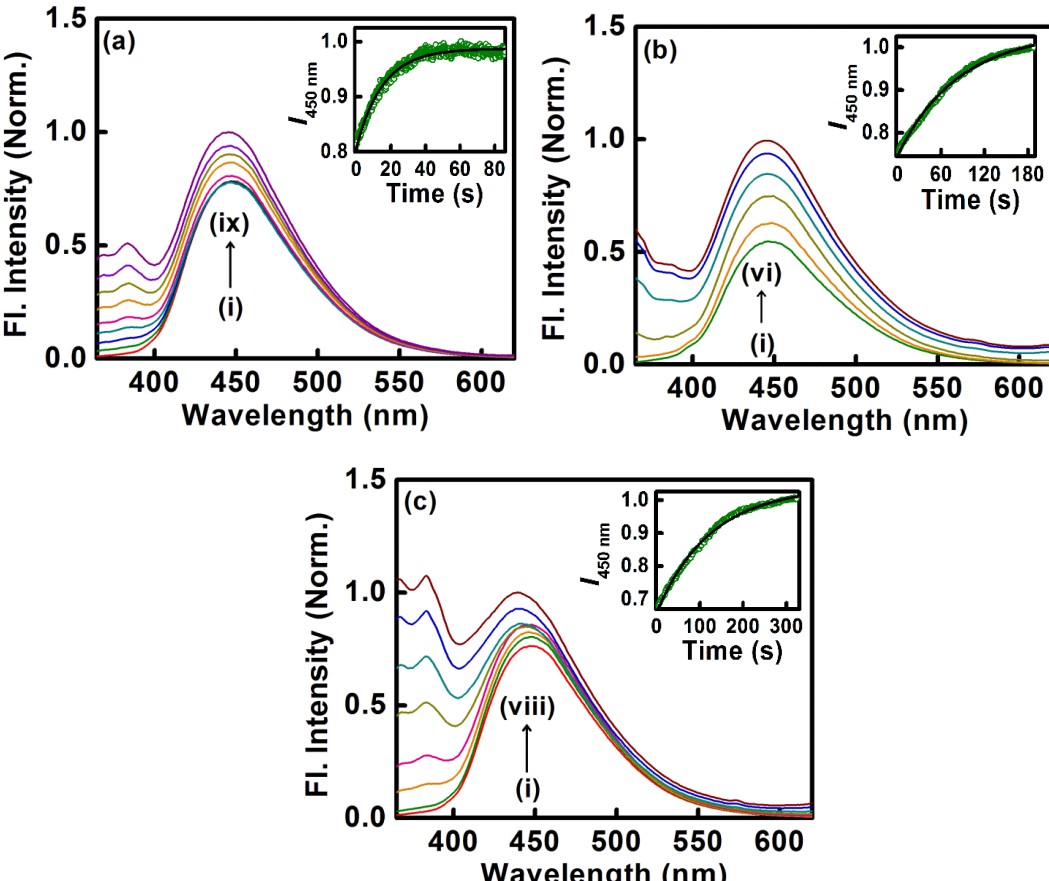

**Figure 3.** Fluorescence spectra ($\lambda_{ex}$ = 348 nm) of the drug NHM (bound to hsDNA) in the presence of increasing concentrations of various cationic surfactants. In panel (**a**): curves (i) → (ix) represent concentrations of CTAB: 0, 50, 100, 150, 200, 250, 300, 350, 400 μM; in panel (**b**): curves (i) → (vi) represent concentrations of TTAB: 0, 20, 40, 60, 100, 120 μM; in panel (**c**): curves (i) → (viii) represent concentrations of DTAB: 0, 50, 100, 150, 200, 250, 300, 350 μM. The insets show the kinetics of the growth of the fluorescence intensity of NHM (bound to hsDNA) following dissociation of the drug induced by the cationic surfactants (monitoring wavelength is $\lambda_{em}$ = 450 nm). The symbols denote the experimental data, and the solid lines denote the fitted lines. The details of the experimental conditions are given in the text. [Reprinted with permission from Ref. [35]. Copyright 2016, Elsevier].

**Table 3.** Summary of thermodynamic parameters for the surfactant-sequestered dissociation of DNA-bound NHM.

| Surfactant [a] | $K_a$ (M$^{-1}$) | $\Delta H$ (kJ mol$^{-1}$) | T$\Delta S$ (kJ mol$^{-1}$) | $\Delta G$ (kJ mol$^{-1}$) |
|---|---|---|---|---|
| CTAB (n = 16) | $(3.5 \pm 0.12) \times 10^5$ | $-3.85 \pm 0.19$ | $27.79 \pm 0.08$ | $-31.63$ |
| TTAB (n = 14) | $(2.6 \pm 0.10) \times 10^5$ | $-1.56 \pm 0.10$ | $29.36 \pm 0.10$ | $-30.93$ |
| DTAB (n = 12) | $(1.84 \pm 0.11) \times 10^4$ | $0.59 \pm 0.10$ | $24.92 \pm 0.11$ | $-24.33$ |

[a] n is no. of carbon atoms in the surfactant tail. [Reprinted with permission from Ref. [35]. Copyright 2016, Elsevier].

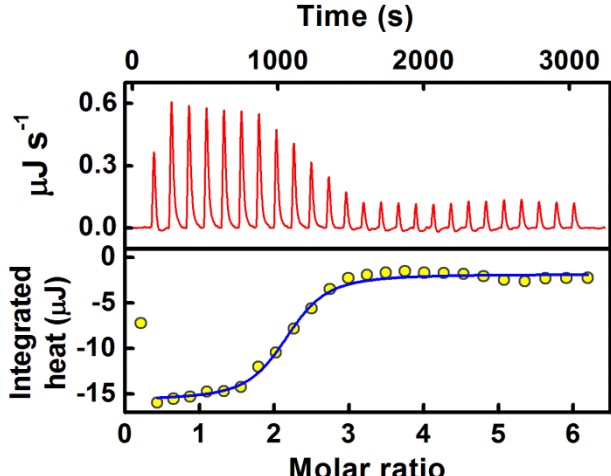

**Figure 4.** The top panel shows the ITC heat burst curves (after correction for the heat of dilution) obtained for the titration of the NHM–hsDNA complex with the cationic surfactant CTAB at 298 K. The bottom panel shows the ITC enthalpogram and the fitted line as obtained after fitting the experimental data to a "single set of sites binding model". [Reprinted with permission from Ref. [35]. Copyright 2016, Elsevier].

### 2.2. Sequestration of Small Drug Molecules from DNA Using Mixed Micelles

Nath and co-workers showed that anionic micelles (SDS micelles) have certain limitations for drug sequestration; for example, the sequestered cationic drug substantially inhabits the surface of micelles and is not encapsulated into the nucleus of micelles due to the high surface charge density, and cationic drug molecules are also unable to gain significant hydrophobic stabilization, etc. [36]. To overcome this challenge, they employed pluronic copolymers in addition to SDS micelles. Such polymer–micelle assemblies form mixed micelles and offer more hydrophobic and electrostatic environments to the bound drugs. These unique features of mixed micelles make them potent candidates for drug sequestration. It has been established that P123-SDS mixed micellar assemblies can successfully deintercalate DNA-bound ethidium bromide; this sequestration may be controlled by the varying chain length of the copolymer used [36,37]. Additionally, cationic surfactant—copolymer mixed micelles, such as P123-CTAB mixed micelles, can be utilized to diminish the denaturing capacity of surfactants on the native structures of biomolecules in the course of their functioning as sequestrating agents for protein-bound drugs [39]. Herein, we discuss the sequestration of an anticancer drug, epirubicin hydrochloride (EPR), from its intercalated complex with well-matched (WM) DNA and a series of mismatched (MM) DNAs, namely cytosine-cytosine mismatched DNA (CC MM DNA), cytosine-thymine mismatched DNA (CT MM DNA), and cytosine-adenine mismatched DNA (CA MM DNA), employing a P123-SDS mixed micellar assembly [38]. EPR exhibits a fluorescence maximum at ~590 nm (excitation at 480 nm). The binding interaction of EPR with DNA is characterized by the quenching of the fluorescence intensity of the drug, while the recovery of the fluorescence intensity of EPR with added SDS to the EPR–DNA (with

WM and MM DNAs) complex containing 0.5 mM P123 is indicative of the release of the bound drug molecules from the DNA scaffolds (Figure 5).

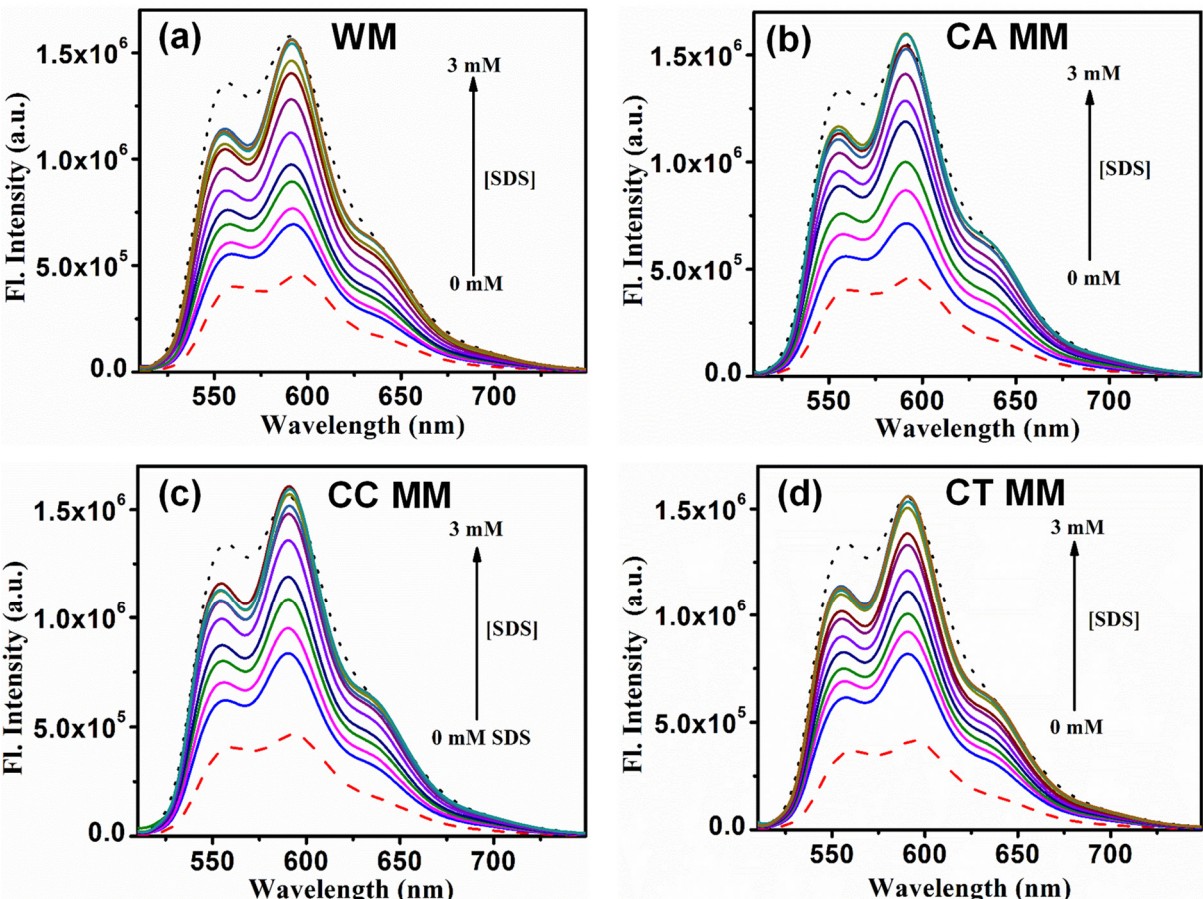

**Figure 5.** Fluorescence emission spectra of 3 μM EPR ($\lambda_{ex}$ = 480 nm and $\lambda_{em}$ = 590 nm) in the EPR–DNA complex containing 0.5 mM P123 in the presence of variant concentrations of added SDS. (**a**) WM DNA, (**b**) CA MM DNA, (**c**) CC MM DNA, (**d**) CT MM DNA. [Adapted with permission from Ref. [38]. Copyright 2021 American Chemical Society].

The deintercalation of the drug from the DNA duplex in the presence of mixed micelles is further affirmed by fluorescence lifetime data, which show that the excited-state fluorescence lifetime of EPR in the EPR–DNA-mixed micelle complex (~1.44 ns) is strikingly close to that of the drug in the EPR–mixed micelle complex (in the absence of DNA) [38]. This result implies that the deintercalated drug molecules are entrapped within the mixed-micellar systems. The release of the drug from DNA with the mixed micelle is also established from other biophysical methods, like fluorescence anisotropy, UV-melting studies, and CD spectral studies. This investigation not only provides detailed insights into the interaction of anticancer drugs with DNA, but also highlights that a mixed-micellar system consisting of an anionic surfactant and neutral triblock copolymer can be employed for detoxification in the case of drug overdose [38].

*2.3. Sequestration of Sanguinarine (SG) from the Liposome Membrane Using Cyclodextrin*

Liposome membranes are self-assembled nanostructures that mimic the lipid bilayer of the plasma membrane. They roll up into a spherical shell having a hydrophilic interior and hydrophobic bilayer region capable of accommodating small amounts of water molecules inside the core and small hydrophobic molecules that can be non-covalently bind the lipid bilayer. These vesicles may comprise phospholipids, non-ionic or ionic surfactants, etc. The size and charge of phospholipid vesicles, known as liposomes, can be tuned by

varying the concentration and composition of the corresponding lipid molecules [73,74]. Over the years, liposomes have captured significant attention in drug delivery research due to their properties, like solubilization, drug loading, and the targeted delivery of drugs [44–46,75,76]. Besides this, Chakraborty and co-workers proposed that liposomes can deintercalate DNA-bound doxorubicin through lipoplex formation [41–43].

In this section, we discuss the sequestration of small drug molecules bound to liposome membranes by cyclodextrin [47]. Cyclodextrins (CDs) are non-toxic, cyclic sugar polymers, soluble in water and in polar solvents, comprised of a structurally well-defined hydrophobic cavity that can accommodate small "guest" molecules mainly via non-covalent interactions. Hence, CDs have been widely applied as "molecular hosts" in a variety of aspects [77–81]. Sanguinarine (SG) is an alkaloid belonging to the quaternary benzo[c]phenanthridine family, having a wide spectrum of therapeutic activities, including antibacterial, antimicrobial, and anti-inflammatory properties [82–85].

The binding interaction of SG with dimyristoyl-L-$\alpha$-phosphatidylglycerol (DMPG) lipid is found to be reflected by a sharp enhancement of the fluorescence intensity of the band distinctive of the iminium form of SG ($\lambda_{em} \sim 580$ nm) together with the diminution of the fluorescence intensity of the band of the alkanolamine form ($\lambda_{em} \sim 418$ nm) with increasing DMPG concentrations (Figure 6a); a schematic representation of the iminium and alkanolamine forms of SG is given in Scheme 2. This observation suggests that a preferential binding interaction of the cationic (iminium) form of the drug with the anionic surface charge of the lipid (DMPG) is assisted by a stabilizing electrostatic interaction [47]. Figure 6b reveals that with added $\beta$-cyclodextrin ($\beta$CD), the fluorescence intensity of the iminium form ($\lambda_{em} \sim 580$ nm) of DMPG-bound SG decreases coupled with an increase in the intensity of the alkanolamine form ($\lambda_{em} \sim 418$ nm). These observations are indicative of the expulsion of the drug molecules bound to the DMPG-lipid with added $\beta$CD [34,86,87]. Such release of the bound drug molecules is also accompanied by a significant reduction in the steady-state fluorescence anisotropy of the DMPG-bound drug in the presence of $\beta$CD, indicating the release of motional restrictions of the bound drug molecules [47]. It is pertinent to note that the complex interaction situation here can lead to several possible equilibria, such as the (i) interaction of DMPG lipid with $\beta$CD leading to disruption of the compact liposome structure, and (ii) inclusion of the drug molecules within $\beta$CD. With a view of the considerably weak interaction of SG with $\beta$CD (association constant in the order of $\sim 10^{-3}$ M$^{-1}$ [82]), the second possibility is rationally discarded [47].

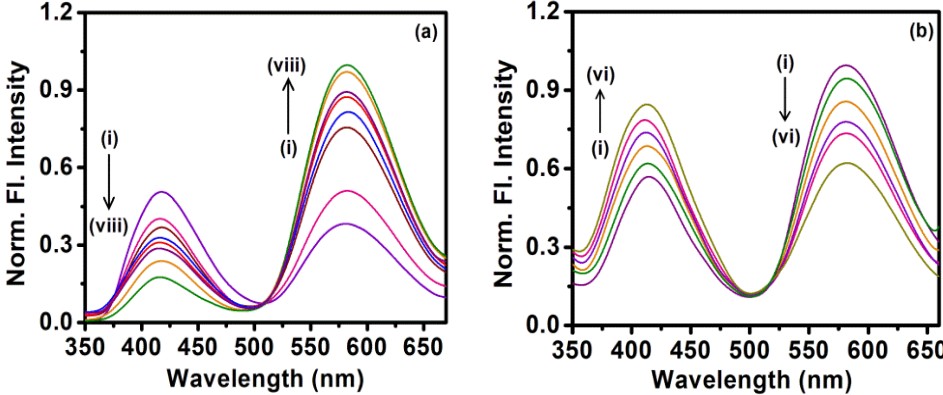

**Figure 6.** (**a**) Fluorescence spectra of the drug SG ($\sim 2.0$ µM) in aqueous buffer (pH 7.4) with added DMPG (the concentration of DMPG increases from (i) to (viii) as 0, 0.05, 0.1, 0.15, 0.3, 0.5, 0.8, and 1.0 mM). (**b**) Fluorescence spectra of the drug DG bound to DMPG with added $\beta$CD (the concentration of $\beta$CD increases from (i) to (vi) as 0, 0.5, 1.0, 1.5, 2.0, and 4.0 mM). The excitation wavelength is 327 nm. [Reprinted with permission from Ref. [47]. Copyright 2018, Elsevier].

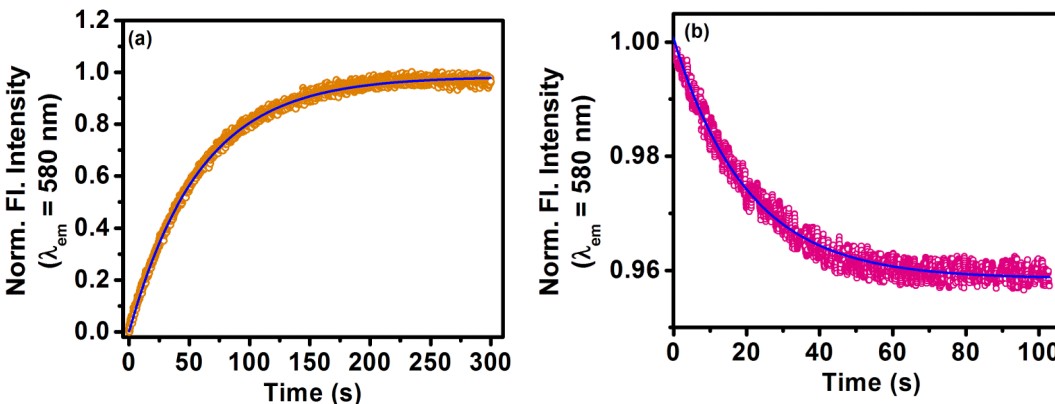

**Scheme 2.** Schematic of chemical structural change of SG depending on the pH of the medium. [Reprinted with permission from Ref. [47]. Copyright 2018, Elsevier].

*Association and dissociation kinetics.* The increase in the fluorescence intensity of SG at $\lambda_{em}$ = 580 nm (Figure 6a) upon interaction with the DMPG lipid has been monitored to study the association kinetics of the SG–DMPG interaction. Figure 7a shows the fluorescence trace describing the kinetics of the association of the SG–DMPG interaction, and the data are fitted to a nonlinear regression as follows:

$$I(t) = \beta e^{-k_a t} + \eta$$

where, I(t) is the variation in fluorescence intensity (at the wavelength under investigation) with time t, with an apparent association rate constant being denoted by $k_a$ ($\beta$ denotes the corresponding amplitude, and $\eta$ is a constant). The interaction of SG with DMPG is found to reveal an apparent association rate constant of $k_a = (1.7 \pm 0.08) \times 10^{-2}$ s$^{-1}$ at T = 303 K [47]. Similarly, the $\beta$CD-induced sequestration of SG from the DMPG-bound state (studied by observing the diminution of fluorescence intensity at $\lambda_{em}$ = 580 nm, Figure 7b) shows an apparent rate constant of dissociation as $k_d = (4.5 \pm 0.18) \times 10^{-2}$ s$^{-1}$ at T = 303 K [47].

**Figure 7.** Normalized fluorescence kinetic traces describing the time evolution of (**a**) the process of the interaction of SG with DMPG as monitored by the growth of the fluorescence intensity of SG at $\lambda_{em}$ = 580 nm and (**b**) the process of the $\beta$CD-induced dissociation of the DMPG-embedded drug as monitored by a decrease in the fluorescence intensity of SG at $\lambda_{em}$ = 580 nm. The scattered symbols represent the raw data, and the solid lines are the fitted curves. The relevant experimental parameters are as follows: [SG] = 2 $\mu$M, [DMPG] = 1 mM and [$\beta$CD] = 8 mM; $\lambda_{ex}$ = 327 nm, T = 303 K, pH ~ 7.4 (aqueous buffer). [Reprinted with permission from Ref. [47]. Copyright 2018, Elsevier].

*Thermodynamics of the DMPG–βCD Interaction.* The thermodynamic parameters for the interaction of the DMPG lipid with $\beta$CD are compiled in Table 4, and the primary heat-burst curves are depicted in Figure 8a. An overall thermodynamically favorable interaction of the DMPG lipid with $\beta$CD is indicated by $\Delta G < 0$, while the process is seen to be increasingly thermodynamically favorable (increasingly negative $\Delta G$) with a rise in

temperature. With an increasing temperature, the interaction is found to be increasingly less exothermic ($\Delta H < 0$), whereas the unfavorable entropic contribution ($T\Delta S < 0$) is found to be progressively less negative (Table 4). The variation in $\Delta H$ with temperature leads to a positive change in the heat capacity, $\Delta C_p = (514 \pm 20)$ J mol$^{-1}$K$^{-1}$, which in turn signifies an instrumental role of hydrophobic hydration in the interaction and is usually connected to the burial of polar surfaces of the interacting moieties (DMPG lipid and βCD) following the interaction [67,69,86–92].

**Table 4.** Thermodynamic parameters for the DMPG–βCD interaction obtained from isothermal titration calorimetry (ITC).

| Temperature (K) | $K_a$ (M$^{-1}$) | $\Delta H$ (kJ mol$^{-1}$) | $T\Delta S$ (kJ mol$^{-1}$) | $\Delta G$ (kJ mol$^{-1}$) | $\Delta C_p$ (J mol$^{-1}$ K$^{-1}$) |
|---|---|---|---|---|---|
| 303 | $(1.08 \pm 0.04) \times 10^3$ | $-31.56 \pm 1.2$ | $-13.95 \pm 0.6$ | $-17.61$ | $514 \pm 20$ |
| 308 | $(1.27 \pm 0.05) \times 10^3$ | $-28.78 \pm 1.2$ | $-10.74 \pm 0.43$ | $-18.04$ | |
| 313 | $(1.47 \pm 0.06) \times 10^3$ | $-26.41 \pm 1.1$ | $-6.91 \pm 0.28$ | $-19.5$ | |

[Reprinted with permission from Ref. [47]. Copyright 2018, Elsevier].

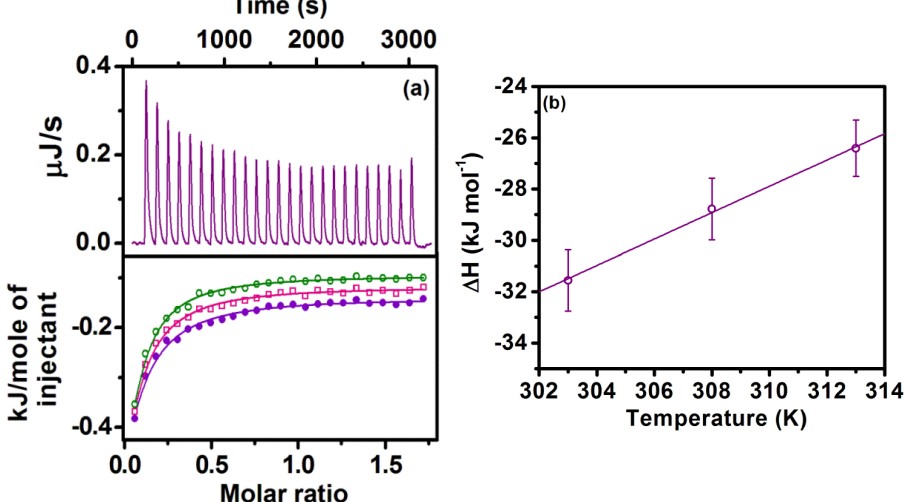

**Figure 8.** (**a**) The top panel shows the ITC heat burst curves (after correction for the heat of dilution) obtained for the titration of the DMPG liposome with βCD at 303 K. The bottom panel shows the ITC enthalpograms and the fitted lines for ITC titrations at 303 K (closed circle), 308 K (open square), and 313 K (open circle). (**b**) Plot of $\Delta H$ vs. T for the DMPG–βCD interaction. [Reprinted with permission from Ref. [47]. Copyright 2018, Elsevier].

***Diffusion of the Drug.*** The translational diffusion coefficients ($D_t$) of SG in various experimental conditions employed in the study, as obtained from fluorescence correlation spectroscopy (FCS,) are summarized in Table 5. The variation in the $D_t$ of SG in different conditions clearly supports the conclusion drawn from spectroscopic and thermodynamic experiments. Hence, an increase in the $D_t$ of the lipid-bound drug with added βCD can be interpreted from the release of the bound drug molecules and thus a release of the motional constraints on the SG molecules with added βCD [86,93–95].

### 2.4. Sequestration of Cryptolepine Hydrate (CRYP) from RNA Using Cucurbit[7]uril

In this section, we discuss the sequestration of an antimalarial drug, cryptolepine hydrate (CRYP) bound to RNA, with the application of host:guest chemistry employing cucurbit[7]uril (CB7) as the host structure [48]. CB7 is well-known to form selective and compact host:guest inclusion complexes with guest molecules [96–101] and thus can be employed as sequestrating agents for small drug molecules bound to various biological receptors, such as RNA [102,103]. The occurrence of the interaction of CRYP with CB7 is evident from the decrease in absorbance (Figure 9a) and the remarkable increase in the

fluorescence intensity (Figure 9b) of the drug with added CB7 [48]. The marked increase in fluorescence intensity may be due to the motional restrictions on the CRYP molecules upon encapsulation in CB7 that results in the lowering of the radiation-less decay channels of CRYP [104]. Figure 9c shows that the absorbance of the drug bound to RNA increases with added CB7, and the absorption spectrum finally resembles that of the interaction of CRYP with CB7 alone. With added CB7, the fluorescence intensity of the RNA-bound drug is found to increase and finally resemble that of the interaction of CRYP with CB7 alone (Figure 9d). Such observations that are qualitatively in a reverse pattern in comparison to those found during the binding of the drug with RNA can be interpreted on the basis of the CB7-induced deintercalation of the bound CRYP molecules from the RNA duplex [104].

**Table 5.** Summary of translational diffusion coefficient ($D_t$) values of SG in various conditions of the experiment.

| System | $D_t$ ($\mu m^2\ s^{-1}$) |
|---|---|
| Aqueous buffer | $63.8 \pm 3$ |
| 1.0 mM DMPG | $9.07 \pm 0.42$ |
| 1.0 mM DMPG + 4.0 mM βCD | $14.33 \pm 0.67$ |
| 1.0 mM DMPG + 6.0 mM βCD | $25.76 \pm 1.21$ |

[Reprinted with permission from Ref. [47]. Copyright 2018, Elsevier].

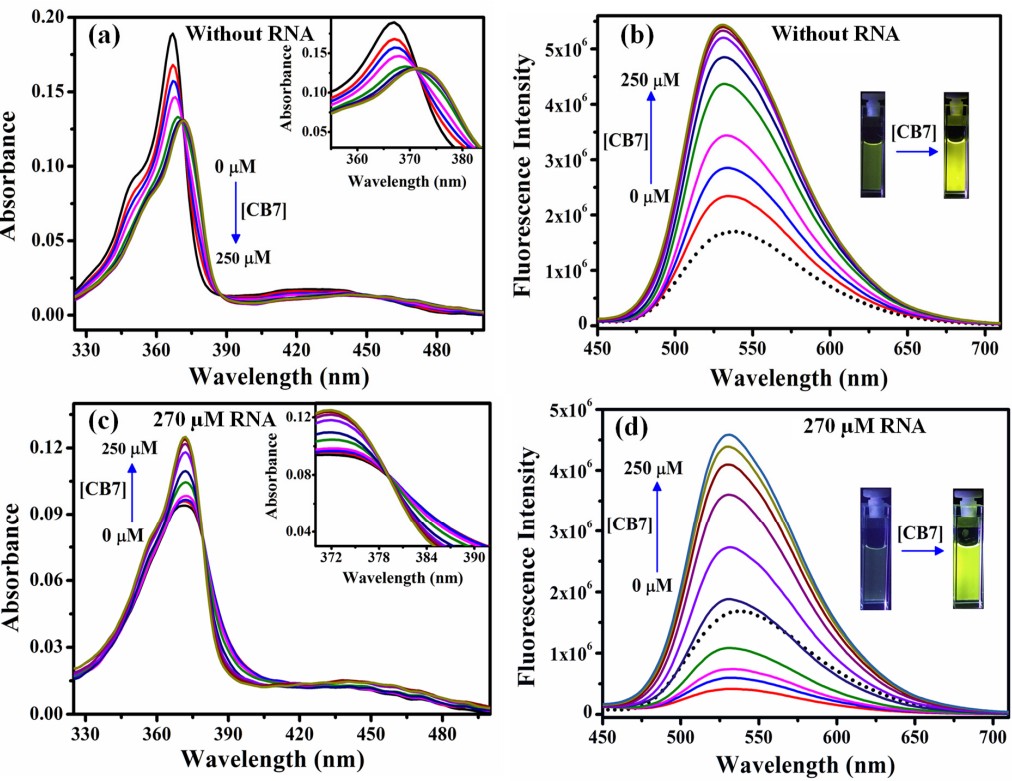

**Figure 9.** (**a**) Absorption and (**b**) fluorescence ($\lambda_{ex}$ = 367 nm) spectra of CRYP with increasing concentrations of CB7. (**c**) Absorption and (**d**) fluorescence spectra of CRYP (bound to RNA) with increasing concentrations of CB7. In panel (**d**), the fluorescence spectrum of CRYP alone in an aqueous medium is represented by the dotted line. The isosbestic points in the absorption spectra are shown in the respective insets. The images of the visible changes in color with the addition of CB7 upon irradiation with ultraviolet light are shown in the respective insets. [Adapted with permission from Ref. [48]. Copyright 2021 American Chemical Society].

### 3. Conclusions

In summary, this review predominantly outlines the in vitro studies on the release of several drugs from biological and bio-replicating systems, with the help of biophysical techniques, such as absorption spectroscopy, fluorescence spectroscopy, and isothermal titration calorimetry. In summary, the results discussed in this article are briefly presented below.

(i) The biological photosensitizer harmane (HM) strongly binds to the DNA duplex principally via the intercalation mode and can subsequently be sequestered using the cationic surfactant CTAB where the cationic surfactant acts as a hydrophobic sink for the drug molecules.

(ii) Another biological photosensitizer, norharmane (NHM), which binds to the DNA duplex, has also been shown to have been sequestered with the aid of cationic surfactants, namely DTAB, TTAB, and CTAB, for which the rate of sequestration of the bound drug is found to be tunable with variations in the chain length of the surfactants.

(iii) The deintercalation of the anticancer drug epirubicin hydrochloride bound to well-matched and mismatched DNA has been shown using mixed micellar assemblies consisting of the anionic surfactant SDS and non-ionic surfactant P123. It was also noticed that the mixed micelles did not alter the native structure of DNA.

(iv) The dissociation of the alkaloid sanguinarine cation bound to negatively charged DMPG liposomes using molecular guest β-cyclodextrin shows the application of host–guest chemistry in this context.

(v) The application of host–guest chemistry in the sequestration of bound drug molecules is further demonstrated in the context of the deintercalation of the antimalarial drug cryptolepine hydrate (CRYP) from the RNA duplex using cucurbit[7]uril hydrate.

Thus, we are optimistic that these results could be helpful for studies on the excretion of drugs from biological environments and for understanding the accompanying mechanisms.

**Author Contributions:** Conceptualization: B.K.P. and S.M., Data collection: R.Y., B.K.P. and S.M., Writing: R.Y., B.K.P. and S.M. All authors have read and agreed to the published version of the manuscript.

**Funding:** CSIR, Government of India (fellowship No: 09/1020(1095)/2020-EMR-I).

**Data Availability Statement:** The data as represented in this review article has been collected from published papers related to the content of the review, duly acknowledging the respective publication.

**Acknowledgments:** R.Y. thanks CSIR, Government of India, for a research fellowship (09/1020(1095)/2020-EMR-I. S.M. thanks IISER Bhopal for providing research infrastructure.

**Conflicts of Interest:** The authors declare no conflict of interest.

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
