# Peer review of "Sequestration of Drugs from Biomolecular and Biomimicking Environments: Spectroscopic and Calorimetric Studies"

_colloids, doi:10.3390/colloids7030051_

Round 1

Reviewer 1 Report

1. Abstract requires the major objective of the review

2. give the detail Sequestration biomolecule in the introduction

3. enlist the drug Sequestration of Drugs from Biomolecular by various researcher

3. use latest refetences

Author Response

Response to Reviewer 1 Comments

Journal: Colloids and Interfaces

Manuscript ID: colloids-2330614

First of all, we sincerely thank the expert reviewer for his/her meticulous inspection of our manuscript and the highly constructive comments and suggestions. We have carefully addressed all the comments of the respected reviewer and have revised the manuscript thoroughly. A point-by-point response to the comments of the reviewer is given below.

In the revised manuscript, all the changes are highlighted in yellow for an easier perusal.

Point 1: Abstract requires the major objective of the review.

Response 1:  We would like to thank the respected reviewer for this suggestion. We substantially modified the abstract of the article, and tried our best to highlight the objectives of the review.

Point 2: Give the detail Sequestration biomolecule in the introduction.

Response 2: We thank the respected reviewer for this suggestion. In the revised version of the manuscript, we have clearly elucidated the mechanism of the drug sequestration from the biomolecules with more relevant examples. In this regard, we have also incorporated few new references in the revised manuscript. Please refer to lines 53 to 95 in the revised version of the manuscript.

“Micelles are self-assemblies of amphiphilic molecules having long hydrophobic tails oriented toward the core and hydrophilic headgroups exposed toward the bulk water. They are thermodynamically stable, nanoscopic dynamic structures typically formed in aqueous medium above a certain concentration (the critical micellar concentration, CMC) and above a certain temperature (the critical micellar temperature (CMT) or the Krafft temperature) [21-23]. Surfactant-assisted dissociation of drug-DNA complex was first reported by Muller and Crothers in 1968 [24]. Later in 2003, Westerlund and co-workers showed that the rate of dissociation of DNA-bound cationic ligands is augmented by the presence of surfactants in both monomeric as well as micellar forms [25]. They studied the binding interaction of four cationic complexes with DNA, suggesting that the intercalative mode of binding to be operational. Further sodium dodecyl sulfate (SDS) surfactant was employed to interact with the cationic complexes by strong electrostatic forces in which the surfactants act as hydrophobic pockets for the sequestration of the complexes. Negatively charged SDS micelles are assumed not to interact with the DNA due to electrostatic repulsion [25]. Furthermore, the same group reported that SDS surfactant drastically accelerates the rate of the dissociation of Ruthenium complexes from calf thymus DNA (ctDNA), with this process being entropically driven [26]. Guchhait and co-workers reported that the interaction of the biological photosensitizer harmane (HM) with herring sperm DNA (hsDNA) is governed by both intercalative and electrostatic forces, and the bound drug was subsequently deintercalated by cationic surfactant cetyltrimethyl ammonium bromide (CTAB) [27]. Patra et. al. reported anionic surfactant SDS micelle-assisted deintercalation of the cationic dye phenosafranin, from DNA [28]. Using Isothermal Titration Calorimetry (ITC), they showed that the overall deintercalation process is enthalpically unfavorable but entropically favorable. Circular dichroism (CD) spectra of SDS with DNA revealed that SDS micelles do not alter the native B-form of DNA [28]. Kundu et. al. explored the cationic CTAB micelle-assisted dissociation of 3-acetyl-4-oxo-6,7dihydro-12H-indolo[2,3-a]-quinolizine (AODIQ) drug from ctDNA. By using spectroscopic and molecular docking studies, they demonstrated that AODIQ binds to DNA at the minor groove. CTAB electrostatically binds to the polyphosphate backbone of DNA and behaves as hydrophobic sink for DNA-bound drug [29]. Apart from cationic and anionic surfactants, Singh and co-workers reported the application of non-ionic surfactant, Triton-X 114 as a sequestrating agent for hs-DNA bound safranin O [30]. Below the CMC, Triton-X 114 is unable to extract the drug molecules from DNA, but beyond its CMC the surfactant is found to extract the DNA-intercalated safranin O. CD spectroscopy further validated that secondary structure of DNA is not influenced by Triton-X 114 at both premicellar and micellar concentrations, thereby making the non-ionic surfactant an effective drug sequestrating probe [30]. Different drug-sequestrating agents for various biomolecular/bio-replicating systems are tabulated in Table 1 [27-46].”

Point 3: Enlist the drug Sequestration of Drugs from Biomolecular by various researchers.

Response 3: As per reviewer’s suggestion, we have enlisted recent literature reports on the various drug sequestrating agents from the biological and biomimicking environments. In this regard, we have also incorporated a table, highlighting the most relevant and recent investigations on the drug sequestration from various biomacromolecules (Table R1). Please refer to Table 1 in the revised version of the manuscript.

Table R1. Different sequestrating agents from different biological and biomimicking environments.

Serial number

Biological/Biomimicking environment

Small Molecule/Drug

Sequestrating Agent

References

1.

Herring Sperm DNA

Harmane

CTAB

27

2.

Calf Thymus DNA

Phenosafranin

SDS

28

3.

Calf Thymus DNA

3-Acetyl-4-oxo-6

,7-dihydro-12H-

indolo-[2,3-a]-

quinolizine

CTAB

29

4.

Herring Sperm DNA

Safranin O

Triton X-114

30

5.

DNA

Mitoxantrone

SDS

31

6.

Heparin

Mallard Blue

SDS

32

7.

Herring Sperm DNA

Norharmane

DTAB, CTAB,TTAB

33

8.

DNA

Ethidium Bromide

P105, F127, and P123 with SDS

34,35

9.

DNA

Epirubicin

P123-SDS Mixed Micelles

36

10.

BSA Protein

CCVJ

CTAB-P123 Mixed Micelles

37

11.

Calf Thymus DNA

Piperine

Surface Active Ionic Liquid

38

12.

Calf Thymus DNA

Doxorubicin

Liposomes

39,40,41

13.

DMPC lipids

Phenosafranin

b-cyclodextrin

42

14.

DMPG Lipids

Nile Red

b-cyclodextrin

43

15.

EYPC Lipids

Phenosafranin, ANS, Nile Red

b-cyclodextrin

44

16.

Liposomes

Sanguinarine

b-cyclodextrin

45

17.

Torula Yeast RNA

Cryptolepine

Cucurbit[7]uril

46

Point 4: Use latest references.

Response 4: We have included recent relevant references in the revised version of the manuscript.

Reviewer 2 Report

The authors presented some examples showing in vitro sequestration of drugs from biomolecular and biomimicking environments. They finally connected the story to modulation of pharmacokinetics by sequestration of drugs from biomolecular and biomimicking environments. It is quite immature  concept. To some extent, it is almost impossible to modulate pharmacokinetics through such kind of complicated process. I can not any evidence to show the potential. 

Author Response

Response to Reviewer 2 Comments

Journal: Colloids and Interfaces

Manuscript ID: colloids-2330614

First of all, we sincerely thank the expert reviewer for his/her meticulous inspection of our manuscript and the highly constructive comments and suggestions. We have carefully addressed all the comments of the respected reviewer and have revised the manuscript thoroughly. A point-by-point response to the comments of the reviewer is given below.

In the revised manuscript, all the changes are highlighted in yellow for an easier perusal.

Point 1: The authors presented some examples showing in vitro sequestration of drugs from biomolecular and biomimicking environments. They finally connected the story to modulation of pharmacokinetics by sequestration of drugs from biomolecular and biomimicking environments. It is quite immature concept. To some extent, it is almost impossible to modulate pharmacokinetics through such kind of complicated process. I can not any evidence to show the potential. 

Response 1: We sincerely thank the expert reviewer for critical inspection of our manuscript and the constructive suggestions. In the revised manuscript we have invested our utmost efforts to work on the issues raised by the revered reviewer. The abstract and conclusion of the manuscript have been rewritten and substantially trimmed down with view to making them consistent with the purpose the manuscript as far as practicable. Several portions within the text of the manuscript have also been rewritten and modified with reference to various literature works along with inclusion of the appropriate places. We have attempted to highlight the scope of the manuscript, that is, a discussion on the application of various molecular assemblies in sequestration of drug molecules bound to biological and/or biomimicking assemblies in in vitro experiments. Given the intrinsic complexities associated with the in-vivo studies to such effects, it is often beneficial to explore the concerned interactions in an in-vitro experimental setup to extract the initial guidelines. The key motivation of the present article is to review the application of various molecular assemblies in the sequestration of drugs bound to different biological assemblies like DNA, RNA, proteins, etc., and biomimicking assemblies like lipid membranes. Herein, we summarize the experimental data showing the extraction of bound drugs by self-assembled molecular aggregates like micelles, mixed micelles, reverse micelles, liposomes, niosomes, etc., as well as by molecular receptors like cyclodextrins, cucurbit[n]urils.

Reviewer 3 Report

This manuscript is a review on sequestration of small molecules biological environment giving examples of receptors (RNA, DNA) using micelles, mixed micelles, cyclodextrins, etc. The review consists of 9 figures and 4 tables obtained from other research.

In general, the manuscript needs extensive correction in scientific writing style, language, organization, and presentation. Please use simplified language and use clear messages. At several places, the redundancy can be seen.

Title: Must be revised, it should provide more details and remove unnecessary keywords. Concise review can be removed.

Abstract:  50% of the initial abstract looks redundant, while the next 50% does not clearly presents the manuscript.

Concussion: It consists of lots of assumptions and redundant info. It must be revised.

I feel that in the present form manuscript will be challenging to understand by most readers, whereas lots of redundant info will confuse the reader. Overall, it lacks proper organization and applications.

It must be thoroughly revised for reconsideration.    

Author Response

Response to Reviewer 3 Comments

Journal: Colloids and Interfaces

Manuscript ID: colloids-2330614

First of all, we sincerely thank the expert reviewer for his/her meticulous inspection of our manuscript and the highly constructive comments and suggestions. We have carefully addressed all the comments of the respected reviewer and have revised the manuscript thoroughly. A point-by-point response to the comments of the reviewer is given below.

In the revised manuscript, all the changes are highlighted in yellow for an easier perusal.

Point 1: This manuscript is a review on sequestration of small molecules biological environment giving examples of receptors (RNA, DNA) using micelles, mixed micelles, cyclodextrins, etc. The review consists of 9 figures and 4 tables obtained from other research.

In general, the manuscript needs extensive correction in scientific writing style, language, organization, and presentation. Please use simplified language and use clear messages. At several places, the redundancy can be seen.

Response 1:  We would like to sincerely thank the revered reviewer for this valuable suggestion. Following the reviewer’s suggestions, we have invested our utmost efforts in revising the manuscript accordingly.

Point 2: Title: Must be revised, it should provide more details and remove unnecessary keywords. Concise review can be removed.

Response 2: As per reviewer’s suggestions, we have modified the title of the manuscript in the revised version and removed unnecessasry keywords.

Point 3: Abstract:  50% of the initial abstract looks redundant, while the next 50% does not clearly presents the manuscript.

Response 3: Following the valuable suggestion of the expert reviewer the abstract has been rewritten and trimmed down substantially.

Point 4: Conclusion: It consists of lots of assumptions and redundant info. It must be revised.

Response 4: Following the valuable suggestion of the expert reviewer, the conclusion has been rewritten highlighting the major points discussed in the text of the manuscript.

Round 2

Reviewer 1 Report

Accept

Author Response

Point 1: Accept.

Response 1: We sincerely thank the respected reviewer for kindly recommending our manuscript for publication.

Reviewer 2 Report

1. I agree that the authors put the effort to revise their manuscript. The key point of this study is sequestration of small drug molecules bound to biomolecules by nanocarriers or nano-assemblies dissolve the issue of drug overdose and drug detoxification.".  In real in vivo conditions, nanocarriers or nano-assemblies should first arrive the site of interest such as different organs/tissues, then have the possibility for sequestration of small drug molecules bound to biomolecules. However, nanocarriers or nano-assemblies themselves will always encounter the big issue of nano-bio interaction. So targeting ability becomes critical for such kind of concept. Please try to discuss (https://doi.org/10.1021/jacs.0c09029; https://doi.org/10.1016/j.addr.2023.114895).

2. Please provide a general cartoon to illustrate the concept of this review.

Author Response

We thank the respected reviewer for his/her close observation of our manuscript and the helpful suggestions/comments. We have addressed the all points raised by the respected reviewer in the revised version of the manuscript. A point-by-point response to the comments of the reviewer is given below.

In the revised manuscript, all the changes are highlighted in yellow for an easier perusal.

Point 1: I agree that the authors put the effort to revise their manuscript. The key point of this study is sequestration of small drug molecules bound to biomolecules by nanocarriers or nano-assemblies dissolve the issue of drug overdose and drug detoxification.".  In real in vivo conditions, nanocarriers or nano-assemblies should first arrive the site of interest such as different organs/tissues, then have the possibility for sequestration of small drug molecules bound to biomolecules. However, nanocarriers or nano-assemblies themselves will always encounter the big issue of nano-bio interaction. So targeting ability becomes critical for such kind of concept. Please try to discuss (https://doi.org/10.1021/jacs.0c09029; https://doi.org/10.1016/j.addr.2023.114895).

Response 1: As per the reviewer’s suggestion, we have now incorporated a few sentences in the revised version of the manuscript. Additionally, we have also included new relevant references in the revised version of the manuscript at appropriate places. Please refer to lines numbers 40 to 52, in the revised version of the manuscript.

“In this context, it may be noted that in the last few years, the evolution in the field of nanomedicine has promised developments of smart systems aiming at in vivo functionality. However, it is important to realize that a large part of the promise developed in this area still relies on rational reasoning rather than concrete experimental evidences, thereby inviting further research to this effect [21,22]. In real in vivo conditions, nanocarriers or nano-assemblies should first arrive at the site of interest such as target organs/tissues, then have the possibility for sequestration of small drug molecules bound to biomolecules. However, nanocarriers or nano-assemblies themselves will always encounter issues related to nano-bio interaction. So, targeting ability becomes critical for these kinds of concepts [21,22]. Naturally, the validation of such concepts still promises ample room for rigorous experiments.”

Point 2: Please provide a general cartoon to illustrate the concept of this review.

Response 2: In the revised version of the manuscript, we introduced a schematic diagram, demonstrating the concept of the review. Please refer to the ‘Graphical Abstract’ section in the revised version of the manuscript.

Reviewer 3 Report

Manuscript is marginally impoved. 

Author Response

Point 1: Manuscript is marginally impoved.

Response 1: We sincerely thank the respected reviewer for appreciating our work.

Round 3

Reviewer 2 Report

The authors addressed the concerns well.